# Synthesis of Manganese Zinc Ferrite Nanoparticles in Medical-Grade Silicone for MRI Applications

**DOI:** 10.3390/ijms24065685

**Published:** 2023-03-16

**Authors:** Joshua A. Stoll, Dorota Lachowicz, Angelika Kmita, Marta Gajewska, Marcin Sikora, Katarzyna Berent, Marek Przybylski, Stephen E. Russek, Zbigniew J. Celinski, Janusz H. Hankiewicz

**Affiliations:** 1Colorado Springs Center for the BioFrontiers Institute, University of Colorado, 1420 Austin Bluffs Pkwy, Colorado Springs, CO 80918, USA; 2Academic Centre for Materials and Nanotechnology, AGH University of Science and Technology, 30-059 Krakow, Poland; 3National Institute of Standards and Technology, 325 Broadway St., Boulder, CO 80305, USA

**Keywords:** silicone, ferrite, nanoparticles, NMR, MRI, MRI-guided surgery

## Abstract

The aim of this project is to fabricate hydrogen-rich silicone doped with magnetic nanoparticles for use as a temperature change indicator in magnetic resonance imaging-guided (MRIg) thermal ablations. To avoid clustering, the particles of mixed MnZn ferrite were synthesized directly in a medical-grade silicone polymer solution. The particles were characterized by transmission electron microscopy, powder X-ray diffraction, soft X-ray absorption spectroscopy, vibrating sample magnetometry, temperature-dependent nuclear magnetic resonance relaxometry (20 °C to 60 °C, at 3.0 T), and magnetic resonance imaging (at 3.0 T). Synthesized nanoparticles were the size of 4.4 nm ± 2.1 nm and exhibited superparamagnetic behavior. Bulk silicone material showed a good shape stability within the study’s temperature range. Embedded nanoparticles did not influence spin–lattice relaxation, but they shorten the longer component of spin–spin nuclear relaxation times of silicone’s protons. However, these protons exhibited an extremely high r_2_^*^ relaxivity (above 1200 L s^−1^ mmol^−1^) due to the presence of particles, with a moderate decrease in the magnetization with temperature. With an increased temperature decrease of r_2_^*^, this ferro–silicone can be potentially used as a temperature indicator in high-temperature MRIg ablations (40 °C to 60 °C).

## 1. Introduction

This paper presents a proof-of-concept synthesis of manganese zinc ferrite nanoparticles in a silicone elastomer matrix. The intended benefit of introducing silicone directly into the reaction mixture is to avoid the practically irreversible formation of nanoparticle clusters. Clustering due to a strong surface interaction is a widely known problem for many different nanoparticle systems [1,2,3,4]. Known methods to break up agglomerates, such as sonication and high-shear mixing, cannot completely disperse particles, although recent electrospray dispersion studies have resulted in varying levels of success [5,6,7]. The most practical way of reducing clustering has been to prevent it from occurring in the first place [8,9,10,11,12]. This is done most frequently by inhibiting direct contact between particles with a surfactant or polymeric coating. This approach is most effective when performed during synthesis because particles never have the opportunity to cluster upon a change in solvent [9].

Silicone (organosiloxane) polymers such as those used in this synthesis are well suited to this approach due to their low reactivity and good heat resistance [13,14]. Despite the fact that silicones and composite materials based on silicone have been successfully used in biomedical devices for many years, materials engineers continue to develop new materials from this group [15,16,17,18]. Synthetic methods for preparing magnetic nanoparticles include electrochemical deposition and/or co-precipitation processes in the pores of silicone, resulting in magnetic elastomers [19]. Coprecipitation has also been used in hydrogels [20]. The work described herein is motivated by the possibility of employing magnetic silicone nanocomposites as temperature sensors during thermal ablation therapy guided by magnetic resonance imaging (MRI). These materials may also improve MR-tagged implants and MR structural phantoms [21,22,23]. 

Thermal ablations locally change the temperature of the targeted tissue to kill cancer cells and to treat other diseases such as arteriovenous malformation and neurological impairment [24,25,26]. High-temperature thermal ablations are performed by heating tissues using laser, radiofrequency (rf) electromagnetic radiation, and ultrasound to a temperature between 50 °C and 80 °C [27]. Low-temperature ablations (cryoablation) are conducted at temperatures between −20 °C and −50 °C or lower [28,29]. Temperature monitoring during surgery is vital to ensure the thorough treatment of affected tissue while minimizing damage to the healthy parts of the organ.

Silicone-based filaments already show promise in monitoring temperature for future cryoablation procedures due to a steady decrease in T_1_ relaxation time across a wide temperature range [30]. The temperature changes in the T_2_ relaxation time are much smaller. Surgeons may prefer, however, to operate viewing T_2_-weighted images as the boundaries of many tissue pathologies are more clearly defined [31]. Note though that T_2_-weighted images typically take a longer time to acquire than T_1_-weighted images. Additionally, as mentioned above, the T_2_ of silicone varies with temperature far less than T_1_, limiting the silicone application of T_2_-based temperature monitoring.

This work attempts to determine the influence of magnetic nanoparticles on the relaxation times of silicones by using, as an example, a silicone elastomer approved by the FDA for medical applications [32]. Previous research on mixed ferrites as temperature-sensitive materials for MRI thermometry has utilized the fact that diluting ferrites with Zn drastically lowers their transition point from a ferrimagnetic to a paramagnetic state [33]. Among MeFe_2_O_4_ ferrites, where Me denotes a 3d metal, MnFe_2_O_4_ ferrite exhibits the highest value of saturation magnetization. Doping with Zn in the range of 0 ≤ x ≤ 0.8 increases the value of saturation magnetization in Mn_1−x_Zn_x_Fe_2_O_4_ ferrite when compared to pure MnFe_2_O_4_ [34]. Moreover, the transition temperature decreases with an increasing Zn content. With a proper Zn concentration, one can lower the transition temperature from 580 K [35], observed in pure MnFe_2_O_4_ ferrite, to the human body temperature. Using an appropriate method for obtaining ferrite nanoparticles allows for synthesizing particles with a specific composition. The selection of the ferrite composition so that the Curie transition is near the body temperature results in a significant change in the magnetic moments of nanoparticles as temperature changes in the region of interest [33]. This, in turn, produces a temperature-dependent alteration in T_2_ and T_2_^*^ of the surrounding silicone. As a result, the brightness in MR images is strongly temperature-dependent, indicating a path for the development of novel composites that can be employed as temperature sensors during MRI-guided ablations.

## 2. Results

### 2.1. Morphological, Structural, and Chemical Characterization

Following the work of Pardo and Lachowicz, the magnetic nanoparticles used in our work were formed by the thermal decomposition of metalloorganic precursors (Fe(acac)_3_, Zn(acac)_2_·xH_2_O, and Mn(acac)_2_) in a silicone polymer matrix [36,37].

Figure 1a shows an example transmission electron microscope (TEM) image of selected nanoparticles synthesized in the presence of a Dow Corning 7-9600 silicone elastomer (DC7). Black spots are the nanoparticles, while the grayish areas are the silicone matrix. Note that the silicone forms a physical barrier which keeps the particles from forming large clusters. Using individual measurements of diameter from nanoparticles on several micrographs, the average diameter was estimated as 4.4 nm ± 2.1 nm (see Figure 1c), and the size distribution was observed to be log normal in character. Although silicone blurs the focus considerably, light spaces between the nanoparticles (observable in Figure 1b) suggest that many are not truly in contact with each other, so that the method was at least partially successful in disrupting agglomeration. Scanning electron microscopy (SEM) imaging of the particles also showed the presence of traces of the silicone layer (see Appendix A).

The crystal structure of nanoparticles grown in silicone was probed using X-ray diffraction; see Figure 2 revealing the spinel structure expected in a magnetic ferrite (JCPDS No. 74-2401). Due to the small sample size and the diluting factor of the silicone, the diffraction peaks were very small (<4000 CPS). Using the Nelson–Riley method, the fitted lattice parameter of 0.8424 nm ± 0.0002 nm was somewhat larger than the bulk lattice parameter, which ranges from 0.8302 nm to 8311 nm [38,39]. 

The peak widths are significantly wider than the instrument baseline. The broadening obscured some peaks, such as (222) and (622), which are, respectively, overlapped by (311) and (533). This peak broadening enabled an estimate to be made of the crystallite size using a Williamson–Hall plot and an assumed shape factor of 0.89 [40,41,42]. The mean crystallite size was determined to be 2.12 nm ± 0.17 nm.

The X-ray spectra gathered using X-ray absorption spectroscopy (XAS) at the L absorption edges of Mn, Fe, and Zn in the total electron yield and partial fluorescence yield are shown in Figure 3. The Mn and Fe L-edge spectra show well-resolved 3d^n^ multiplet structures at two absorption edges with the L3 at a 12–15 eV lower energy than the L2 edge (around 640 eV and 652 eV for Mn and around 710 eV and 722 eV for Fe). Multiplets related to a 2p→3d excitation are followed by a gradual and less structured increase in intensity, known as an edge step (2p→4s excitation). In the case of Zn, no multiplets are observed due to its full 3d^10^ electronic configuration, thus only the edge step is present. The characteristic multiplet features, corresponding to the different charges and crystal field symmetries of 3d ions, were analyzed by comparing them with the spectra of other reference oxides probed at the XAS station of Solaris [43].

The Mn L_3_-edge spectrum of a ferrite is dominated by a high peak at 640.2 eV that corresponds to Mn^2+^ situated in octahedral (O_h_) sites. However, other features, denoted on Figure 3a with dashed green lines, differ in intensity with respect to a MnO standard, which are attributed to Mn^3+^ in tetrahedral (T_d_) sites. The lower intensity of the main peak at 640.2 eV relative to the rest of the Mn L_23_ multiplet features supports the interpretation of a significant fraction of manganese ions in higher valence states (3+ or even 4+). A similar shift in the proportion of peak intensity was also observed between the iron L-edge spectra of ferrite and an α-Fe_2_O_3_ reference. The lack of a characteristic minimum between the main L_3_ multiplet features is attributed to a significant fraction of Fe^2+^ ions in octahedral sites, which is related to the distribution of zinc among O_h_ and T_d_ sites. Studied nanoparticles seem to resemble the normal bulk spinel more closely, having more zinc in T_d_ sites.

The quality of the fluorescence spectra (see Figure 3b) is insufficient to analyze their shape in detail since the differences observed are within statistical noise. Nevertheless, data from the measured samples exhibited features similar to those observed in the mixed ferrite. A qualitative comparison between chemical compositions may be made by comparing the relative increase in the fluorescence intensity on all the edges. However, since the X-ray beam size is rather large, artifacts related to the density and thickness of the studied and reference samples, as well as absorption and fluorescent scattering by silicone, may strongly affect these numbers.

The composition of nanoparticles grown in silicone was determined using the results of inductively coupled plasma mass spectroscopy (ICP-MS) and assuming a formula of (Mn_x_Zn_1−x_)_y_Fe_3-y_O_4_ based on the metal content since oxygen cannot be measured using this method. The determined composition of Mn_0.55_Zn_0.55_Fe_1.9_O_4_ is near the target composition. The over-doping of the ferrite, with the ratio of y to 3-y of not being 1 to 2, is due to the original proportion of iron to other metals in the reaction mixture. In addition, EDX and XPS studies of the obtained particles were performed, which also confirm the results of the above studies (see Appendix A).

### 2.2. Silicone Characterization

The temperature stability of the Dow Corning 7-9600 bulk silicone material is critical for reliable MRI experiments. To determine the stability of the shape, pure silicone samples were cured in 5 mm glass NMR tubes and exposed to low and high temperatures. Three samples were cooled from room temperature down to −20 °C, and three were heated up to 60 °C. The cooled samples showed clear defects in the form of long voids that appeared below 10 °C. The heated samples remained clear and did not show any visible structural changes. To retain the integrity of the samples, NMR and MRI experiments were conducted above 10 °C. 

The infrared absorption profile of the ferrite is simple and consists of a peak at 540 cm^−1^ (see Figure 4). It is attributed to the overlapping absorbances of octahedral and tetrahedral *E* symmetry vibrations of metal–oxygen bonds and is consistent with the energies previously reported for MnZn ferrites [44]. They are slightly higher than bulk ferrite with a comparable composition, which may be attributed to lattice strain [45]. The *T*_2_ symmetry phonon mode is expected to have a peak circa 400 cm^−1^, but this is beyond the edge of the detection range. The broad peaks at 1550 and 1400 cm^−1^ are attributable to acetylacetonate residues that were left over from the reaction.

Importantly, the doped silicone sample shows all peaks of the DC7 silicone matrix intact. The absorbance at 2961 cm^−1^ is due to methyl C-H stretching. There is a slight diminution of this peak, but it cannot be quantified without an accurate template for subtracting the nanoparticle absorbance. Silicones are far more thermally stable under nitrogen than under oxygen; however, in conjunction with thermogravimetric data, this could indicate the onset of depolymerization at 280 °C [46,47]. There is also slight chemical shifting. The Si-O-Si stretching peak at 1007 cm^−1^ is chemically shifted to 1015 cm^−1^ and the Si-C stretching peak at 786 cm^−1^ is shifted to 795 cm^−1^, likely due to steric hinderance in the vicinity of the particles. These shifts involving the polymer backbone, rather than functional groups, suggest a nonspecific mode of surface adsorption [48]. Any polymerization caused by the procedure cannot be determined as the only change in the spectrum of DC7 upon hydrolyzation is a minor peak, not visible on the data presented in Figure 4, forming at 2850 cm^−1^. It is caused by the stretching of the H-C bond adjacent to the newly formed C-Si bond. Due to the low intensity of the absorbance pattern of the silicone covering the nanoparticles, this would be undetectable above the background noise.

In addition, thermogravimetric analysis (TGA) was completed in an argon atmosphere to determine the percentage of the polymer layer in the total mass of the studied system. The TGA curve for silicone with embedded MnZn ferrite nanoparticles is shown in Figure 5. The initial decrease in weight, between 100 and 200 °C, can be associated with the loss of residual water that is incorporated in the silicone layer. The decomposition of the polymeric layer starts above 200 °C, which is common for polydimethylsiloxanes. The majority of the degradation occurs between 300 °C and 700 °C by self-reaction in the absence of oxygen to form small cyclic dimethylsiloxanes, which completely evaporated [49]. The total weight loss for the sample was calculated as 59.23%, indicating that in the tested system, the share of nanoparticles was about 40% (see Figure 5).

### 2.3. Magnetometry

Figure 6 shows the temperature dependence of the mass magnetization and hysteresis loop of grown nanoparticles measured at 200 K. The magnetic hysteresis loop shows a negligible remanence and very small values of a coercive field (see Figure 6a), indicating that the particles exhibit superparamagnetic behavior at this temperature. The mass magnetization (Figure 6b) decreases moderately with an increasing temperature.

### 2.4. Nuclear Magnetic Resonance (NMR) Measurements

The ratio of integrals of the NMR spectra of the protons in DC7 silicone to water was compared at 20 °C. The obtained value of 0.70 corresponds well to previously reported numbers in other polymers (0.65–0.80) [30]. The high abundance of protons indicated by this value means the silicone will appear bright in proton-weighted MRI scans with parameters set for water. The summary of the temperature studies of nuclear relaxation is presented in Figure 7. As seen in Figure 7a, even a low concentration of nanoparticles drastically reduces the T_2_^*^ obtained from the observed linewidth. By comparison, adding nanoparticles to silicone has little effect on the spin–lattice relaxation times (Figure 7b). Spin echo amplitudes from the Carr–Purcell–Meiboom–Gill sequence used for measurements of the T_2_ relaxation clearly show multiexponential decay [50,51]. These measurements were analyzed with a two-exponential decay model. Long and short components of T_2_ were denoted as a and b, respectively, as shown in Figure 7c. Doping silicone with nanoparticles has no effect on a short component; however, it significantly shortens a long component.

### 2.5. Magnetic Resonance Imaging

MRI measurements were conducted at 18 °C (magnet’s bore temperature) on phantoms consisting of pure silicone and silicone doped with 0.6 mmol L^−1^ and 1.0 mmol L^−1^ of mixed MnZn ferrite. Two groups of experiments were conducted: (1) an attempt to measure the apparent diffusion coefficient (ADC) and (2) the comparison of morphological image quality obtained with spin-echo multi-slice (SEMS) and gradient-echo multi-slice (GEMS). 

The value of ADC for the studied silicone is well below the lower limit of 10^−12^ m^2^s^−1^ in the MRI scanner configuration. Using the scanner diffusivity limit, a 4 nm MnZn ferrite particle diameter (approximated as spherical), and an 8 × 10^5^ Am^−1^ magnetic moment, a scaling parameter of $\Delta \omega \tau_{D}>180\gg 1$ was obtained. In the inequality, Δω describes the Larmor frequency shift of silicon protons due to the magnetic field created by a particle at its equator, and $\tau_{D}$ is the translational diffusion time of the silicone protons. For details on the model of the particles’ motional averaging regimes, and the related calculations, see the paper by Voung et al. [52]. $\Delta \omega \tau_{D}\gg 1$ clearly implies that the system is out of the motional averaging regime. That is, the loss of phase coherence among nuclear spins happens faster than the diffusion responsible for the spin phase averaging. Note that this is similar to situations where there is fast diffusion, but the particle sizes are on the order of micrometers [53]. 

Cross-sectional MR images of the phantom containing pure DC7 silicone, silicone doped with two concentrations of MnZn ferrite nanoparticles, and the reference sample of deionized water with NiCl_2_ is shown in Figure 8. For the comparison of proton-weighted MRI and T_2_^*^-weighted MRI, the images were acquired with spin-echo and gradient-echo sequences, respectively. One can appreciate almost the perfect rephasing of nuclear spins by a standard two radio-frequency pulse spin echo sequence in all samples, including the samples doped with ferrite nanoparticles (Figure 8b). Because of a long repetition time (TR) of 8 s (TR > 5T_1_ in DC7 samples) and a relatively short echo time (TE) of 10.37 ms, the intensity of all samples in Figure 8b is proportional to the amount of hydrogen atoms in the samples. The ratio of intensity in the region of interest marked by red circles in Figure 8a of pure DC7 to the water sample is 0.65 ± 0.07, which is slightly smaller than expected from the NMR spectra calculations, but within the experimental error. As the gradient echo sequence inherently does not refocus the local magnetic field inhomogeneity images of the DC7 samples doped with magnetic particles, they are heavily T_2_^*^-weighted, as seen in Figure 8c. This unique property of the gradient echo sequence is the foundation of the use of magnetic particles as a temperature contrast agent for MRI [54]. Due to the short T_2_^*^ of the 1.0 mmol L^−1^ sample (see Figure 7a), the signal is within the noise and the sample image is not visible. We note that the traces of the signal are visible for the very short echo time of TE < 2 ms.

## 3. Discussion

The structure of nanoparticles produced in the silicone-based synthesis is that of mixed spinel, as indicated by X-ray diffraction and XAS. The extreme width of the XRD peaks is indicative of very small particle sizes. Oxygen vacancies in the structure could not be detected by XRD, but the possible presence of Mn^4+^ ions, as signaled by XAS, suggests that stoichiometric quantities of oxygen would tend to neutralize the charge and favor the Mn^2+^ state. The incorporation of Mn into octahedral sites is likely encouraged by the slight deficit of Fe, as indicated by ICP-MS.

These results show that the synthesis of ferrites in silicone produces the expected crystallographic spinel phase typically obtained by thermal degradation syntheses. We expect that this will be the general rule for silicones, so long as reactive alkenyl groups are excluded from the reaction mixture. Future syntheses should reduce the temperature the mixture is held in. The silicone laden with nanoparticles may be added in small quantities to a silicone mixture prior to curing to augment the hydrogen T_2_^*^ relaxation of the silicone. Silicones augmented in this way may be useful as MRI phantoms. 

The nanoparticles exhibited a linear decrease in mass magnetization values with an increasing temperature over the entire measured temperature range (230 K to 350 K). The measured rate of change in the magnetization with temperature was 0.078 Am^2^kg^−1^K^−1^. We note that a steep decrease in the moment is desirable for tMRI applications. 

It has long been known that relaxation mechanisms which affect T_1_ must also affect T_2,_ but the converse does not apply [55]. Thus, an observed reduction in T_2_ values but not in T_1_ is unusual, but not impossible. The effective T_2_ of pure silicone is a composite of two distinct relaxation rates. A rapid relaxation, which can be attributed to the rotation of monomer units, and a slow relaxation, that is ascribed to the vacillation of polymer chains [56,57]. It is interesting to note that doping silicone with magnetic particles shortens the long component of T_2_ (almost 40% at 60 °C), while the short component is largely unaffected by doping. 

The dynamics of the particles’ magnetic moment was insufficient to have much influence on the T_1_ relaxation of the silicone protons, but promoted the dephasing of the spins, noticeably decreasing T_2_^*^. As we see in Figure 7a, the values of T_2_^*^ are drastically short. If converted to relaxivity r_2_^*^, as shown in Figure 9, its values above1200 L s^−1^ mmol^−1^ are much higher than that of maghemite-based human serum albumin hybrid nanoparticles, (482 L s^−1^ mmol^−1^ reported) which are generally considered powerful r_2_^*^ contrast agents [58].

The authors believe, however, that such a high value of r_2_^*^ is not solely the effect of the presence of nanoparticles, but it appears only when combined with the lack of proton diffusion in silicone. Hypothetically, the 3 T field locks the magnetic moments of these superparamagnetic nanoparticles into alignment (see Figure 6a) and without the proton diffusion, conditions similar to the static dephasing regime predominate [51,59]. The full explanation of the observed phenomenon requires a comparison of these presented results in silicone to results obtained from matrices with a certain degree of proton diffusion, such as hydrogels, and is the subject of ongoing study.

As a general comment, systematic studies of ferrite nanoparticle growth in silicone matrices and the influence they may have on all relaxation times are necessary prior to the application of these composites during MRI-guided ablations. Such studies should include the effect particle size and concentrations have on silicone’s T_1_, T_2_, and T_2_^*^ relaxation times. Moreover, the magnetic heating contribution by nanoparticles should be minimized. Superparamagnetic particles exhibit very small hysteretic behavior, which significantly limits the sources of heating caused by the fast switching field gradients of the MRI scanner [60,61]. However, this hysteretic behavior strongly depends on nanoparticle size.

## 4. Materials and Methods

### 4.1. Synthesis of Nanoparticles in Silicone

A 7-9600 Soft Filling Elastomer was purchased from Dow (Midland, MI, USA). Chloroform (≥99%), squalene, dibenzyl ether (DBE, ≥98.0% GC), zinc acetylacetonate hydrate (Zn(acac)_2_·xH_2_O), manganese(II) acetylacetonate (Mn(acac)_2_, 97%), and iron(III) acetylacetonate (Fe(acac)_3_, 97%) were purchased from Sigma Aldrich (St. Louis, MO, USA). Anhydrous ethyl alcohol (ethanol, 99.8%) and n-hexane were purchased from POCH/Avantor (Gliwice, Poland). Acetone was purchased from Chempur (Piekary Śląskie, Poland). Deionized water was used for all aqueous solutions.

The metalloorganic precursors were added in the following amounts: 427.4 mg (1.2 mmol L^−1^) of Fe(acac)_3_, 103.2 mg (0.4 mmol L^−1^) of Zn(acac)_2_·xH_2_O, 101.2 mg (0.4 mmol L^−1^) of Mn(acac)_2_ to a 250 mL 3-necked flask containing a stir bar. Then, 20 mL of Dow-Corning 7-9600 soft filling elastomer part A was added. Part A was selected because it contains the catalyst and not the unsaturated alkenyl groups. Therefore, it is less likely to polymerize during synthesis. The flask was capped with septa and a 0.5-meter-high distilling column with argon bubbled through at approx. 500 mL/min. The mixture was then heated to 130 °C. At this temperature, 10 mL of dibenzyl ether (DBE) was added to the mixture to limit the vaporization of the metal salts. The mixture was heated first to 200 °C and held for 30 min. Next, its temperature was increased to 280 °C and maintained at this temperature for 60 min to drive the process to completion. 

The tendency of many ethers to form explosive peroxides is well documented [62]. It is advisable to use fresh DBE, especially given the high temperatures required for synthesis. Because DBE will reflux during synthesis, thereby increasing the possibility of peroxide formation, it is important that oxygen be displaced from the reactor prior to heating and to maintain the resulting solution under an inert atmosphere until it is fully cooled. The use of silicone-based septa or other high-temperature polymers is important for protecting the solution during the reaction.

After cooling down to room temperature, the solution was then washed with 300 mL of equal parts acetone and n-hexane and precipitated for 3 days in the field of a strong magnet. The precipitate was resuspended in chloroform, drawn down again using a strong magnet, and then resuspended a final time in chloroform. 

### 4.2. Microscopy Methods

Transmission electron microscopy (TEM) was performed using a Tecnai TF 20 X-TWIN microscope (FEI, Hilsboro, OR, USA) upon samples dispersed with ethanol on a copper grid. Energy dispersion X-ray spectroscopy confirmed the presence of all the desired elements. The clustering behavior of the particles was probed using a FEI VERSA 3D scanning electron microscope (SEM) (FEI, Hilsboro, OR, USA) or a Tescan VEGA3 SEM (Brno, Czech Republic). The SEM and TEM images were analyzed with Image-J software (version 1.53, NIH, Bethesda, MD, USA) by individually measuring the long axis of the nanoparticles from several micrographs.

### 4.3. X-ray Diffractometry

Powder X-ray diffraction (XRD) was performed with a Rigaku SmartLab multipurpose X-ray system (Tokyo, Japan) equipped with a Cu X-ray source. Diffractograms were analyzed with Rigaku PDXL 2 (version 13, Rigaku, Tokyo, Japan) data processing software and the particle sizes were determined, based on XRD measurements, using Mathematica 13 (version 13.1, Wolfram Research, Champaign, IL, USA).

### 4.4. Spectroscopy

Soft X-ray absorption spectroscopy (XAS) was performed at room temperature at the PEEM/XAS beamline station of the Solaris National Synchrotron Radiation Center in Krakow, Poland [43]. The beamline was passed through a plane grating monochromator which has a resolving power of E/ΔE > 4000. Samples of pure DC7 and DC7 doped with nanoparticles were probed at the Mn, Zn, and Fe L2 edges using the surface-sensitive (~5 nm information depth) total electron yield (TEY) and volume sensitive partial fluorescence yield (PFY) detection modes. The sample spectra were compared with that of a MnZn ferrite of a known composition. 

Fourier transform infrared spectroscopy (FTIR) was performed using a Nicolet iS5 (Waltham, MA, USA) equipped with an iD7 ATR diamond lens for attenuated total reflectance over the 400 to 4000 cm^−1^ (12 to 120 THz) range. Peaks were identified using *Origin 2022 software* (version 2022b, OriginLab Corporation, Northampton, MA, USA).

### 4.5. Magnetometry

Magnetometry was performed using a vibrating sample magnetometer (Model 7407, by Lake Shore, Carson, CA, USA) using a 1.5 T applied field. Hysteresis curves were gathered at 200 K and the temperature dependence of the magnetic moment was measured from 230 K to 350 K.

### 4.6. Nuclear Relaxation Time Measurements 

A portion of the nanoparticle suspension in chloroform was dried in open air to leave a residue of nanoparticles which was then thoroughly mixed into additional DC7 Part A to make a 2.0 mmol L^−1^ (by formula weight) suspension. This was divided into 3 portions, 2 of which were further diluted to make 1.0 mmol L^−1^ and 0.6 mmol L^−1^ suspensions. DC7 Part B was added to each and mixed thoroughly. The samples were vacuum degassed under a residual pressure of 120 mbar until no bubbles could be seen on the surface. A syringe with a 5 mm bore was used to inject a metered amount of material into standard 5 mm glass NMR tubes.

The temperature dependence of spin–lattice (T_1_), spin–spin (T_2_), and spin dephasing (T_2_^*^) relaxation times of protons in the DC7 elastomer in a pure sample and a sample doped with MnZn ferrite nanoparticles at a concentration of 0.6 mmol L^−1^ (160 μg/mL) and 1 mmol L^−1^ (240 μg mL^−1^) was determined in the temperature range of 20 °C to 60 °C, in 10 degree steps. Measurements were conducted using a 3.0 T (128 MHz) pulse NMR spectrometer (Redstone console manufactured by Tecmag, Houston, TX, USA) and a standard bore 54 mm superconducting magnet manufactured by Oxford Instruments, Abingdon, UK [63]. The samples were cured in standard 5 mm glass tubes (Wilmad-LabGlass, Vineland, NJ, USA).

A 3.0 T NMR magnet was initially shimmed using a pure DC7 sample at 20 °C to achieve the spectra linewidth below 4 Hz. Linewidth was defined as the full width at half maximum of a Lorentzian line fit to the experimental spectra obtained after a Fourier Transform of the free induction decay (sequence details: 90° pulse 13.4 μs, TR = 5 s). The observed linewidth at each temperature was used to calculate the corresponding T_2_^*^ values using: $T_{2}^{*}=\frac{1}{\pi \cdot FWHM}$ [64]. The spectrum, T_1_ (125 ms) and T_2_ (90 ms), of the sample of 10 mM of NiCl_2_ deionized water was also registered at 20 °C. The spectra of protons in water and pure DC7 were integrated to obtain the ratios of hydrogen atoms in silicone to those in water. T_1_ relaxation times were measured using a standard inversion recovery sequence (180° pulse = 26.8 μs, 90° pulse = 13.4 μs, repetition time = 5 s; inversion time array included 20 delays in the range of 20 ms to 14 s). T_2_ relaxation times were measured using a Carr–Purcel–Meiboom–Gill sequence (90° pulse = 13.4 μs, 180° pulse = 26.8 μs CPMG loop consisting of 20 delays from 60 ms to 1240 ms).

### 4.7. Magnetic Resonance Imaging

The MRI experiments were carried out in an Agilent preclinical scanner with a 3.0 T, 30 cm bore magnet (Agilent, Santa Clara, CA, USA). For details, see reference [65]. Images were taken at the magnet’s bore temperature of 18 °C using the samples from NMR arranged in a custom polylactic acid sample holder. Two groups of experiments were conducted: (1) an attempt to measure the apparent diffusion coefficient (ADC) and (2) a comparison of the image intensity and quality obtained with spin-echo (SEMS) and gradient-echo (GEMS). 

ADC was measured with a standard pulsed-gradient spin-echo (PGSE) sequence with b-values reaching to 2000 s/mm^2^ [66]. The imaging geometry parameters for SEMS and GEMS were identical: field of view = 25 × 25 mm^2^, slice thickness = 5 mm, acquisition matrix = 64×64 pixels, and in-plane resolution = 0.39 mm/pixel. Sequence timing parameters: for SEMS—TR = 8 s, TE = 10.37 ms; for GEMS—TR = 8 s, TE = 2.36 ms, flip angle = 90°.

The NMR spectroscopic data, relaxation data, and MR images were processed using Python-based software developed in-house. Figures were created using Origin 2022 (version 2022b). The statistical analysis of regression and correlation was conducted using the Prizm software (GraphPad Prism version 5.00 for Windows, GraphPad Software, San Diego, CA, USA). Mathematica 13 (version 13.1).

### 4.8. Inductively Coupled Plasma-Mass Spectrometry

The atomic ratios of metal ions present in the samples were determined by ICP-MS. The samples were dissolved in fresh aqua regia, the solutions were diluted, and then the pH was neutralized. These samples were then tested according to EPA protocol 200.7 by the Colorado Department of Public Health and Environment Laboratory in Denver, CO, USA [67].

## 5. Conclusions

The proposed synthesis method allows mixed MnZn ferrite nanoparticles to grow in a medical-grade silicone matrix. The grown particles are well dispersed, superparamagnetic, and show high magnetization values. Due to the restricted diffusion of silicone molecules, embedded particles do not influence the T_1_ and T_2_ relaxation times of silicone ^1^H nuclei, but drastically reduce their T_2_^*^ values. This phenomenon is clearly visible in the intensity of T_2_^*^-weighted MR images acquired with the gradient echo method. However, the observed moderate temperature dependence of magnetization requires a further tuning of the composition of nanoparticles for the use of fabricated magnetosilicones as temperature sensors for MRI thermometry.

## Figures and Tables

**Figure 1 ijms-24-05685-f001:**
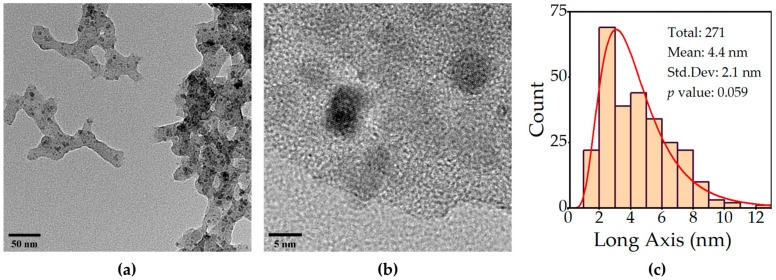
Morphology analysis of nanoparticles grown in silicone. (**a**) Representative TEM image. (**b**) High-resolution TEM image of the nanoparticles. (**c**) Histogram of particle sizes fitted to a lognormal size distribution.

**Figure 2 ijms-24-05685-f002:**
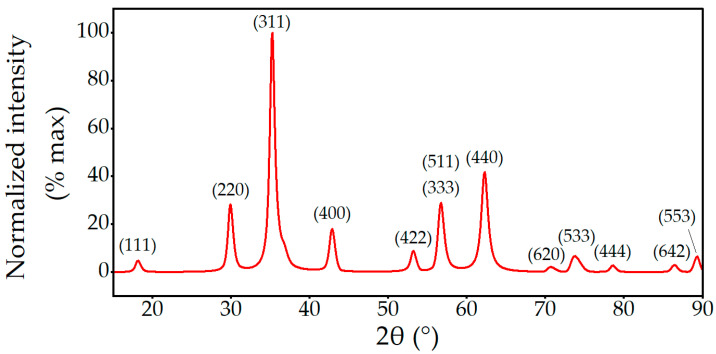
Background subtracted X-ray diffractogram of silicone-coated nanoparticles using Cu radiation. Labeled peaks are characteristic of a spinel structure with significant broadening due to particle sizes.

**Figure 3 ijms-24-05685-f003:**
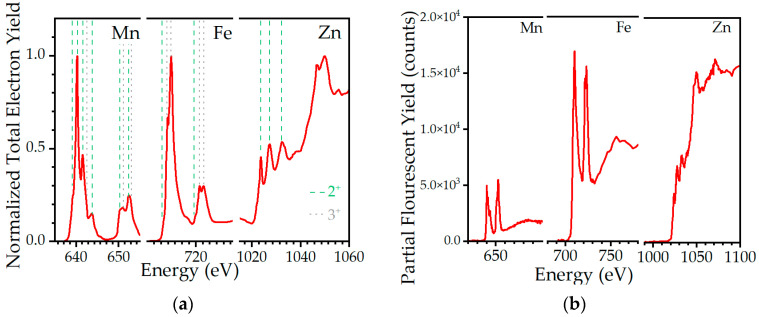
X-ray absorption spectra of nanoparticles grown in silicone. (**a**) The total electron yield spectra in the Mn, Fe, and Zn L absorption range. Green dashed lines indicate features characteristic of ions with 2+ oxidation states and gray dotted lines indicate features of 3+ oxidation. (**b**) The measured fluorescence yield of Mn, Fe, and Zn.

**Figure 4 ijms-24-05685-f004:**
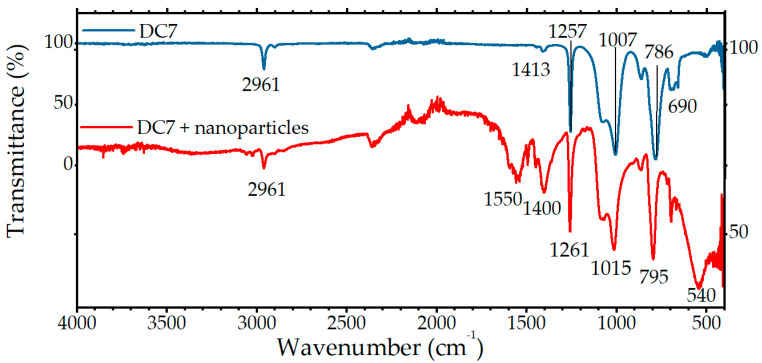
FTIR absorption spectra of pure DC7 (top blue) and the DC7 used in the synthesis of MnZn ferrite nanoparticles (bottom red) clearly showing peaks associated with its matrix.

**Figure 5 ijms-24-05685-f005:**
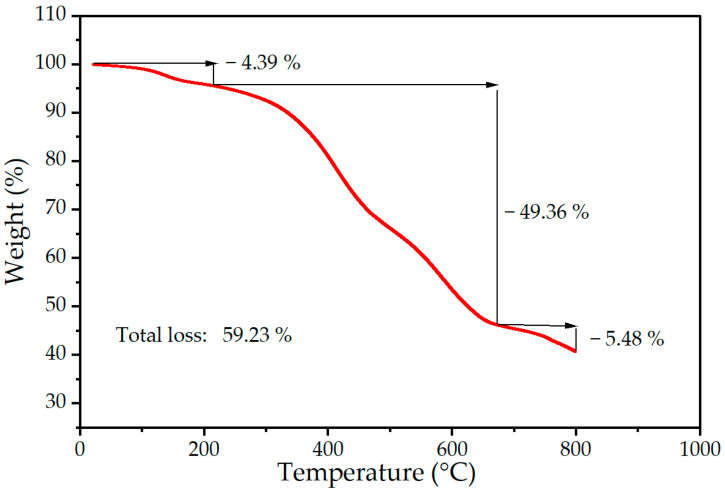
Thermogravimetric analysis for the determination of nanoparticle weight in the silicone matrix.

**Figure 6 ijms-24-05685-f006:**
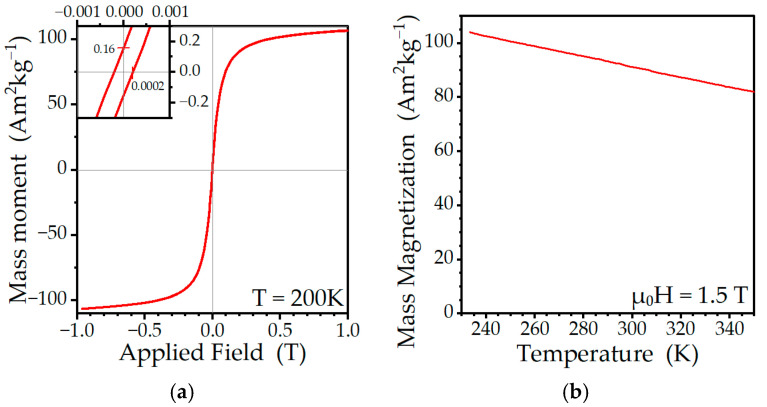
Magnetization study of MnZn ferrite nanoparticles grown in silicone. (**a**) The hysteresis loop measured at 200 K. (**b**) The temperature dependence of the mass magnetization measured in a 1.5 T field.

**Figure 7 ijms-24-05685-f007:**
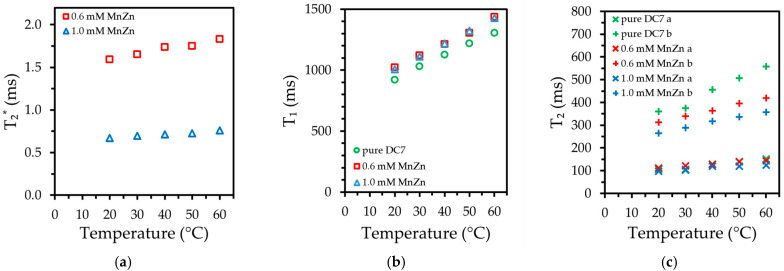
Temperature dependence of nuclear relaxation times of silicone protons at 3.0 T. (**a**) spin dephasing time T_2_^*^, (**b**) spin–lattice time T_1_, (**c**) spin–spin time T_2_.

**Figure 8 ijms-24-05685-f008:**
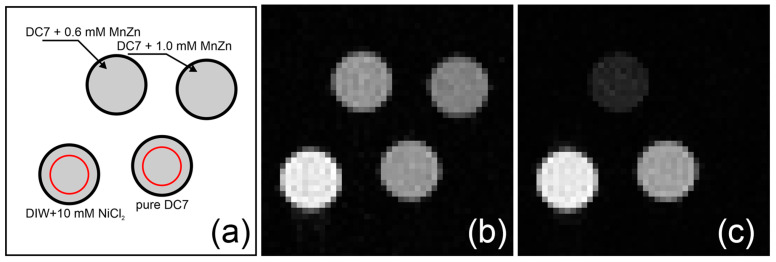
Cross-sectional MR images of the phantom at 3.0 T (18 °C). (**a**) Diagram showing the sample arrangement. Red circles denote the regions used to calculate the ratio between the intensity belonging to pure DC7 and deionized water doped with NCl_2_ (10 mmol L^−1^). (**b**) SEMS image, (**c**) GEMS image. For imaging parameters see the Methods section.

**Figure 9 ijms-24-05685-f009:**
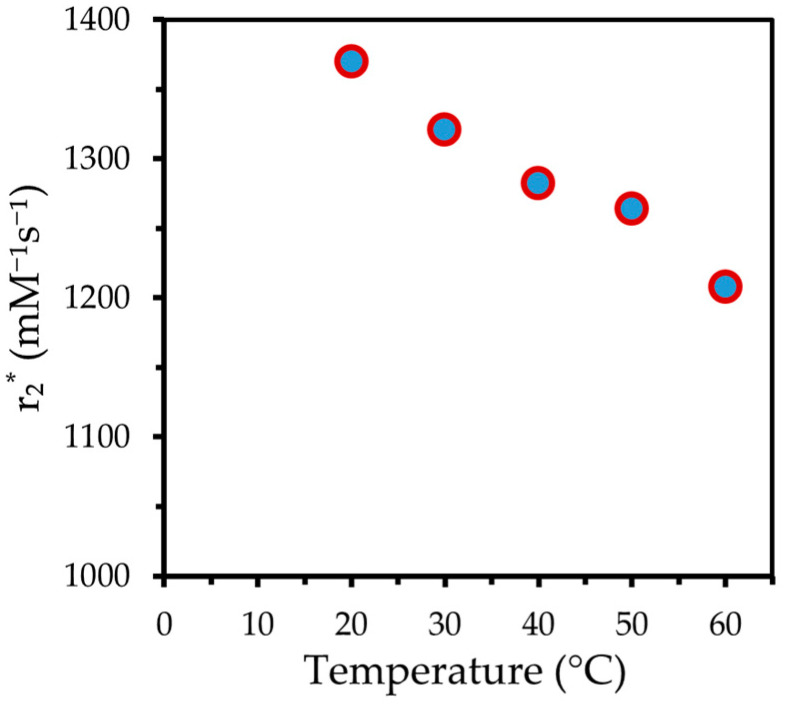
Temperature dependence of r_2_* relaxivity for protons in silicone doped with MnZn nanoparticles.

## Data Availability

The data presented in this article is available upon request to the corresponding author.

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
