# Peer review of "Synthesis of Manganese Zinc Ferrite Nanoparticles in Medical-Grade Silicone for MRI Applications"

_ijms, 2023, doi:10.3390/ijms24065685_

Round 1

Reviewer 1 Report

NO COMMENT

Author Response

We greatly appreciate the reviewers' efforts to carefully review the paper.

Reviewer 2 Report

Thank you for the opportunity to review the work "Synthesis of manganese zinc ferrite nanoparticles in medical grade silicone for MRI applications."

I congratulate the authors for the excellent work and the clarity presented in the manuscript.

Below I make my considerations so that the manuscript can be considered for publication in this journal.

The work proposes to produce a hydrogen-rich silicone composite embedded with magnetic ferrite nanoparticles for use as an indicator of temperature change in magnetic resonance-guided thermal ablations (MRIg).

For this, the authors use a route based on the thermal decomposition of ferrite precursors in the continuous silicone medium to control the size and avoid agglomeration of the MNPs. In the sequence, the authors describe the physical-chemical, and magnetic characterizations of the materials obtained and conclude that the characteristics of this composite make it promising as a temperature sensor in the range of 40 to 60°C.

Regarding the Introduction: The authors explain well the motivation of the work.

However, I suggest the authors clarify why zinc ferrite was studied explicitly about other ferrites.

I would also like to suggest that the authors discuss the importance of the size of these MNPs for this application. Since they discuss the importance of the materials being well dispersed and then justify the synthetic route used precisely to obtain small and non-agglomerated materials, it would be interesting to address the importance of the size of the MNPs in this type of application.

Results:

Figure 1 could be discarded as it does not add much to work.

On line 82, "Note that the silicone forms a physical barrier which keeps the particles from forming large clusters." Based on the micrograph shown in Figure 1(a), this sentence is not good, as clusters are pretty obvious, especially in the most concentrated regions of MNPs. To confirm such a statement, a micrograph with a higher resolution would be necessary to show the absence of such agglomerates clearly.

On line 83, "Using a lognormal distribution model, we estimate the average diameter is 4.4 ± 2.1 nm (see Figure 2b)". If the MNPs were counted, why do the authors mention that the average diameter was estimated through a fitting? Furthermore, I believe that the information on the number of particles, standard deviation, and R2 of the fitting could be added as complementary information in Figure 2(b).

At line 88, "see supplementary Figure S1", No file containing supplementary manuscript information could be accessed.

In line 91, "0.8424 ± 0.0002 nm," does the precision of 0.002 nm have any physical significance?

On line 136, "The over-doping of the ferrite, with the ratio of y to 3-y of not being 1 to 2, is due to the original proportion of iron to other metals in the reaction mixtures." Why was the ratio of Iron used not 1 to 2, as is commonly used in ferrite syntheses?

On line 148, I believe the authors could add a little information making it clear that absorption refers to Metal-Oxygen stretching.

In line 160, the authors mention the presence of residues. In this context aiming at the intensity of the bands, it is assumed that the amount of residue is high. In this context, how does this affect the properties of the MNPs? And what is the advantage of this route compared to coprecipitation, which is a cheap synthesis without contaminants?

In figure 4, no information is given about the band at 2961 cm-1, which almost ceases to exist with the addition of MNPs, why?

On line 176, "This can be easily explained by the difference in the relative weight percentage of polymer to the inorganic core in the system." The statement assumes that the values obtained are obvious, but this is not so clear. For example, why is this mass loss so small? Is the amount of MNPs that high in this composite? In fact, could the authors better inform the concentration of MNPs in the composite? A figure of the dispersion obtained would be much more enlightening than Figure 1 used.

In Figure 6(a), It would be interesting to zoom in on the central region to make it clear that there is no hysteresis.

Discussions:

The first part could be omitted or reduced since it only describes the already known LaMer model.

On line 270, the information referring to the XRD cannot be verified as it is not possible to access the supplementary material.

In line 276,  "Our results show that the incorporation of silicone preserves the chemistry of the product ferrites expected from typical thermal degradation syntheses", it was not clear what the authors mean by "preserves the chemistry", since the MNPs present the surface contaminated with acac residues and stoichiometry is not 1:2, I can't understand what the authors are referring to.

In line 280, the authors comment on the bands that would indicate the adsorption of silicone on the surface of the MNPs; it would be interesting to identify the next band (~720 cm-1), which has a much more significant change in its profile and could indicate some interaction of the MNPs with the middle.

Materials and Methods

I would only put the name of the software used in italics.

In line 401, "The tendency of many ethers to form explosive peroxides is well documented" - I would like to congratulate the authors for mentioning such information in the work, I think it is of paramount importance to avoid accidents with new researchers in the area.

In the end, the authors do not present a conclusion section, although they are not obliged. In my view, I believe that it would be an important section to close all the important findings of the submitted manuscript.

In general, it is an excellent work, with very clear results and discussions, I hope that my suggestions can help in some reflections for the authors to improve even more the quality of the manuscript.

And I believe that after that, the work can be considered for publication in this journal.

Reviewer 3 Report

1. please give the DPI for the DLS.

2. How about the SEM for the sample characterization, it could be provided.

3. “This approach is most effective when performed during synthesis because particles never have the opportunity to cluster upon change of solvent. Some of the updated refs could be highlighted, such as Inorg. Chim, Acta 546(2023)121297 and Colloid Surface B, 2022, 213, 112432.

4. The authors should list a Table for comparision on the MRI application between the similar materials and the title material.

Reviewer 4 Report

This work attempts to determine the influence of magnetic nanoparticles on the relaxation times of silicones by using, as an example, a silicone elastomer approved by the FDA for medical applications. I found the manuscript clearly written and easily understandable. It contains useful information which may be interesting to some of the Int. J. Mol. Sci. readers. Therefore, I recommend the publication of the paper after addressing the following comments:

The structure of the materials lacks enough characterizations. XPS analysis should be carried out to determine the elemental composition of the samples and the oxidation state of each element. SEM and EDS should be reported to investigate the morphology of manganese zinc ferrite nanoparticles. Presenting high-quality HRTEM images would provide additional insights regarding the morphological and structural properties of the synthesized materials. 

Results and discussion need improvement. It should be presented with clarity and precision and should be explained by referring to the literature and should interpret the findings in view of the results obtained. 

JCPDS No. should be provided for XRD measurements. The particles sizes should be calculated by the Debye Scherrer formula citing the following papers:

Ultrasonics sonochemistry 55 (2019): 44-56. DOI: 10.1016/j.ultsonch.2019.03.001

Journal of environmental management 267 (2020): 110629. DOI: 10.1016/j.jenvman.2020.110629

There are many short subsections in the materials and methods section, which could be merged.

Some references (especially those cited in the introduction part) should be replaced with recently published papers.
